# FREE LUNCH AT INFERENCE: TEST-TIME REFINEMENT FOR DIFFUSION MODELS

## ABSTRACT

Diffusion probabilistic models (DPMs) have recently achieved state-of-the-art performance in generative tasks, surpassing traditional approaches such as GANs and VAEs in both sample quality and training stability. Despite their success, DPMs suffer from high computational cost and slow sampling, since they require sequential denoising across many timesteps. Existing acceleration methods primarily focus on reformulating the reverse process as an ODE/SDE and applying advanced numerical solvers. While effective, these approaches largely overlook the geometric properties inherently induced by the Gaussian process in DPMs. In this work, we investigate the geometric behavior of DPMs in the latent variable manifold, revealing an overlooked isotropic property derived from their Gaussian formulation. Building on this characteristic, we introduce a lightweight test-time refinement that can be seamlessly embedded into existing samplers. Our method reduces the discretization error of sequential sampling methods and accelerates the convergence of parallel sampling strategies, without requiring extra training or additional model evaluations. Extensive experiments across multiple datasets demonstrate that our approach consistently improves both generation quality and efficiency, while remaining fully compatible with existing methods. By uncovering and exploiting the isotropic nature of DPMs, this work provides a new perspective on the geometric foundations of DPMs and offers a complementary direction for advancing their efficiency. As a snapshot result, when integrated into UniPC, our method improves the FID score on LSUN bedroom from 39.89 to 20.08 with 4 function evaluations.

## 1 INTRODUCTION

Diffusion Probabilistic Models (DPMs) have rapidly become a dominant paradigm for high fidelity data generation Sohl-Dickstein et al. (2015); Ho et al. (2020), following their introduction into the field of generative modeling. Compared to conventional approaches such as Generative Adversarial Networks (GANs) Goodfellow et al. (2014) and Variational Autoencoders (VAEs) Kingma & Welling (2013), DPMs offer several advantages that they are more robust during training, produce samples of superior quality, and provide a controllable generation process. These strengths have enabled DPMs to be successfully deployed in diverse applications, including image synthesis Song et al. (2021); Ho et al. (2020), path planning Yu et al. (2024); Ren et al. (2025), and other complex generative tasks Chen et al. (2024b).

Despite these advantages, DPMs remain computationally intensive, incurring long generation times due to the large number of sequential denoising steps (*i.e.*, many function evaluations) Ho et al. (2020). Unlike one-shot generators such as GANs Goodfellow et al. (2014), sampling from DPMs requires sequentially evaluating the model across many timesteps. This computational bottleneck originates from the Gaussian forward process assumed in DPMs, which in turn requires an iterative reverse process to progressively denoise a signal from pure Gaussian noise back to data Song et al. (2021). As a result, the deployment of DPMs is limited on computationally constrained platforms, restricting their broader adoption.

To reduce this cost, existing research has primarily focused on reformulating the diffusion process in terms of ordinary or stochastic differential equations (ODE/SDE) Song et al. (2020; 2021). This connection enables the use of advanced numerical solvers in ODE/SDE to accelerate sampling *e.g.*,

high-order solvers Lu et al. (2022); Liu et al. (2022); Zhang & Chen (2023), and Picard iterations are introduced to parallel the reverse process Shih et al. (2023). While these approaches substantially reduce generation time, most prior work has focused primarily on solver design.

In contrast, comparatively less attention has been paid to the geometric characteristics of DPMs. Understanding and exploiting the geometry of DPMs offers a complementary avenue for improving both efficiency and fidelity of DPMs. A small but growing number of works have begun to explore geometric structures in DPMs. Prior studies impose geometric perspectives in time space to reformulate trajectories, regularize training, or redesign noise schedules for improved stability and fidelity Chen et al. (2024a); Song et al. (2023); Karras et al. (2022); Karczewski et al. (2025). For example, flattening trajectories and carefully designing noise schedules have been shown to accelerate the reverse process by reducing the number of timesteps required in the sampling process Song et al. (2023); Karras et al. (2022), while recent geometric on spacetime manifold approaches speed up sampling by approximating geodesic flows on learned manifolds Karczewski et al. (2025).

Complementary to ODE/SDE solvers and time space geometric approaches (*e.g.*, flatten trajectories, *etc.*), this work adopts a distinct geometric viewpoint on the latent-variable manifold in DPMs. In particular, we identify an isotropic structure induced by the Gaussian process underlying these models. Building on this insight, we propose a **zero-cost**, **test-time** refinement that integrates seamlessly with existing sampling frameworks. The plug-in refinement reduces discretization error in sequential sampling methods and accelerates the convergence of Picard iterations in parallel samplers, without any additional model evaluations or training.

Our key contributions can be summarized as follows:

- Geometric perspective on latent manifold of DPMs. We reveal and formalize an isotropic property inherent in the Gaussian process of DPMs, providing a new geometric viewpoint on their latent manifold.

- Lightweight zero-cost, test-time refinement. Leveraging this property, we design a zero-cost, test-time refinement method that can be seamlessly integrated into existing solvers, reducing the discretization error of sequential sampling methods without extra training or additional model evaluations.

- Faster parallel sampling. We demonstrate that our approach accelerates the convergence of Picard-iteration–based parallel samplers via a dual-update mechanism, improving both efficiency and image quality.

- Experiments across multiple datasets and baselines demonstrate consistent gains in both sample quality and generation speed, highlighting the effectiveness and generalization of our method. As a snapshot result, when integrated into UniPC, our method reduces the Fréchet inception distance (FID) score on LSUN bedroom from 39.89 to 20.08 with 4 function evaluations.

> We believe that uncovering and systematically leveraging the isotropic structure inherent in the Gaussian process underlying DPMs can pave the way for future advances in diffusion modeling.

## 2 RELATED WORK

The practical deployment of diffusion probabilistic models (DPMs) is often limited by their computational expense, as the reverse process requires many sequential function evaluations and consequently long runtimes. Following the introduction of denoising diffusion probabilistic models (DDPM) Ho et al. (2020), efforts to accelerate sampling emerged almost immediately. DDIM Song et al. (2021) addressed this by breaking the Markov chain in the reverse process, enabling large sampling step sizes and substantially reducing the number of iterations needed for image generation. It also introduced a quadratic, non-uniform timestep schedule that further mitigates discretization error. In addition, DDIM highlighted a connection between discrete-time denoising processing and continuous-time ordinary differential equations (ODE) Song et al. (2021). At the same time, related work established a link between DDPM and stochastic differential equations (SDE) Song et al. (2020). This ODE/SDE viewpoint opens the door to high-order solvers that control discretization

error at large step sizes without increasing the number of function evaluations Karras et al. (2022); Dormand & Prince (1980). Pseudo-Numerical Methods for Diffusion Models (PNDM) Liu et al. (2022) propose a pseudo-function to approximate the behavior of differential equations. Based on this perspective, DPM-Solver Lu et al. (2022) and its variants Lu et al. (2023); Zheng et al. (2023) employ high-order exponential integrators tailored to DPMs, and DEIS Zhang & Chen (2023) similarly applies an exponential integrator to accelerate sampling.

Classic ODE correction techniques have been adapted to diffusion sampling to refine predictor updates and improve image quality Zhao et al. (2023); Xue et al. (2024). Complementary scheduling strategies include adaptive step-size control based on scaled error estimates to balance local errors across steps Jolicoeur-Martineau et al. (2021), as well as auxiliary networks that predict timesteps during sampling Zhou et al. (2024).

Large sampling steps inherently increase discretization error, motivating knowledge distillation approaches that transfer the behavior of a high-fidelity teacher requiring many timesteps to a student model that achieves comparable performance with fewer steps Salimans & Ho (2022); Berthelot et al. (2023); Song et al. (2023). Despite reducing the number of inference steps, distillation methods require additional training and thus introduce substantial cost overhead.

A small but growing body of work explores geometric structure in diffusion, shaping trajectories and schedules in time–state space to improve stability and fidelity Chen et al. (2024a); Song et al. (2023); Karras et al. (2022); Karczewski et al. (2025). For example, trajectory flattening and schedule redesign can shorten the effective integration path and better align the reverse dynamics with the target distribution Song et al. (2023); Karras et al. (2022), while recent perspectives accelerate sampling by approximately following geodesic flows on learned manifolds Karczewski et al. (2025).

To exploit hardware efficient utilization, Shih et al. (2023); Lu et al. (2025) proposed a parallel sampling method based on Picard iteration, enabling concurrent computation across timesteps. Although this approach can yield speedups, it often requires additional model evaluations to ensure convergence, and the overall runtime depends on the convergence rate of iterations.

Complementary to existing methods, we adopt a latent manifold perspective. We identify an isotropic structure induced by the Gaussian process and leverage it to derive a zero-cost, test-time refinement that integrates seamlessly with existing sampling methods. The proposed plug-in method reduces discretization error in sequential sampling methods and accelerates the convergence of Picard iterations in parallel sampling methods. Importantly, our method does not require any additional model evaluations or training.

# 3 PRELIMINARY

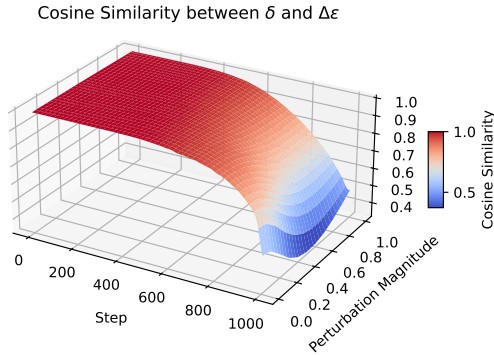

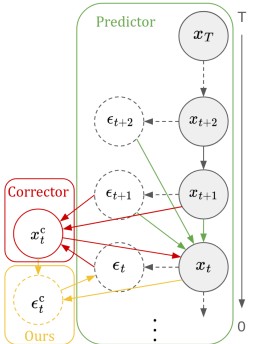

Figure 1: Cosine similarity between the input perturbation $\delta$ and the resulting change in predicted noise $\Delta\epsilon$ for stable diffusion v1-4. $\Delta\epsilon$ and $\delta$ are shown in Equation (8). High values indicate strong alignment between input changes and output changes.

Figure 2: Predictor-corrector procedure, corrector refines latent variables, ours refines predicted noise

**Diffusion Probabilistic Model.** Let $q_{\text{data}}(\boldsymbol{x})$ denote the data distribution. DPMs assume a forward Gaussian process, where at $t = 0$ we have $q_0(\boldsymbol{x}_0) = q_{\text{data}}(\boldsymbol{x})$ with $\boldsymbol{x}_0 = \boldsymbol{x}$ being the clean samples.

The forward process is defined as a Gaussian transition Lu et al. (2022):

$$q_{0t}(\boldsymbol{x}_t \mid \boldsymbol{x}_0) \;=\; \mathcal{N}\big(\boldsymbol{x}_t \mid \alpha_t \boldsymbol{x}_0, \, \sigma_t^2 \mathbf{I}\big), \tag{1}$$

where $t \in [0, T]$, and $\alpha_t, \sigma_t$ form the noise schedule. This schedule is designed such that the marginal distribution at the terminal time satisfies $p(\boldsymbol{x}_T) \approx \mathcal{N}(\mathbf{0}, \tilde{\sigma}^2 \mathbf{I})$ for $T > 0$ and $\tilde{\sigma} > 0$. Moreover, the signal-to-noise ratio $\frac{\alpha_t^2}{\sigma_t^2}$ decreases monotonically with respect to $t$, ensuring the Gaussian process Lu et al. (2022).

Furthermore, the forward process shares the same transition with stochastic differential equation (SDE) Kingma et al. (2021):

$$\mathrm{d}\boldsymbol{x}_t = f(t)\boldsymbol{x}_t \, \mathrm{d}t + g(t) \, \mathrm{d}\boldsymbol{w}_t, \boldsymbol{x}_0 \sim q_0(\boldsymbol{x}_0), \tag{2}$$

where $\boldsymbol{w}_t \in \mathbb{R}^d$ is the standard Wiener process, $f(t) = \frac{\mathrm{d}log\alpha_t}{\mathrm{d}t}$ and $g^2(t) = \frac{\mathrm{d}\sigma_t^2}{\mathrm{d}t} - 2\frac{\mathrm{d}\log\alpha_t}{\mathrm{d}t}\sigma_t^2$.

The corresponding reverse process of the forward diffusion from timestep $T$ to $0$, are given by Song et al. (2021):

$$\mathrm{d}\boldsymbol{x}_t = [f(t)\boldsymbol{x}_t - g^2(t)\nabla_{\boldsymbol{x}} \log q_t(\boldsymbol{x}_t)] \, \mathrm{d}t + g(t) \, \mathrm{d}\bar{\boldsymbol{w}}_t, \tag{3}$$

where $\bar{\boldsymbol{w}}_t$ denotes a standard Wiener process in reverse time and $p_T(\boldsymbol{x}_T) \approx \mathcal{N}(\mathbf{0}, \tilde{\sigma}^2 \mathbf{I})$.

The training objective of DPMs is to approximate the scaled score function $-\sigma_t \nabla_{\boldsymbol{x}} \log q_t(\boldsymbol{x}_t)$ with a neural network $\boldsymbol{\epsilon}_\theta(\boldsymbol{x}_t, t)$ parameterized by $\theta$. For a well-trained network,

$$\boldsymbol{\epsilon}_\theta(\boldsymbol{x}_t, t) \;\approx\; -\sigma_t \nabla_{\boldsymbol{x}} \log q_t(\boldsymbol{x}_t). \tag{4}$$

**The Reverse Process.** In contrast to the forward process, which drives samples away from the data manifold, the reverse process removes noise to trace a trajectory back toward it by solving SDE/ODE

$$\frac{\mathrm{d}\boldsymbol{x}_t}{\mathrm{d}t} = f(t)\boldsymbol{x}_t + \frac{g^2(t)}{2\sigma_t^2}\boldsymbol{\epsilon}_\theta(\boldsymbol{x}_t, t), \boldsymbol{x}_T \sim \mathcal{N}(\mathbf{0}, \tilde{\sigma}^2 \mathbf{I}), \tag{5}$$

where $f(t) = \frac{\mathrm{d}log\alpha_t}{\mathrm{d}t}$ and $g^2(t) = \frac{\mathrm{d}\sigma_t^2}{\mathrm{d}t} - 2\frac{\mathrm{d}\log\alpha_t}{\mathrm{d}t}\sigma_t^2$.

For a given latent variable $\boldsymbol{x}_s$ at time $s$, using the half log-SNR($\lambda$) replace the $\alpha$ and $\sigma$, $\lambda_t = log(\frac{\alpha_t}{\sigma_t})$. The analytical solution for latent variable at time $t$ for a given latent variable at time $s$ is Lu et al. (2022); Zhao et al. (2023)

$$\boldsymbol{x}_t = \frac{\alpha_t}{\alpha_s}\boldsymbol{x}_s - \alpha_t \int_{\lambda_s}^{\lambda_t} e^{-\lambda} \boldsymbol{\epsilon}_\theta(\boldsymbol{x}_\lambda, \lambda) d\lambda. \tag{6}$$

ODE/SDE solvers serve as the Predictor, approximating the integral in Equation (6) via a truncated series expansion, which introduces local discretization error. To mitigate this, a Corrector is applied at each step to refine the predicted latent state, as shown in Figure 2 Zheng et al. (2023). The Predictor and Corrector can be any ODE/SDE solver:

$$\boldsymbol{x}_t \leftarrow \text{Predictor}\big(\boldsymbol{x}_s, \boldsymbol{\epsilon}_\theta(\boldsymbol{x}_s, s), Q\big), \quad \boldsymbol{x}_t^c \leftarrow \text{Corrector}\big(\boldsymbol{x}_s, \boldsymbol{\epsilon}_\theta(\boldsymbol{x}_t, t), Q\big), \tag{7}$$

where $Q$ is the query of previously predicted noise. Leveraging intrinsic characteristics of DPMs, our method provides an extra refinement step on the predicted noise.

## 4 METHODOLOGY

### 4.1 OVERLOOKED CHARACTERISTIC OF DIFFUSION MODELS

Existing training-free methods primarily focus on improving ODE/SDE solvers to reduce sampling time or enhance image quality. However, these approaches often overlook a key geometric property induced by the Gaussian process in DPMs that the variations in the model's output are inherently aligned with variations in its input. In this section, we formalize this property and demonstrate that it arises directly from the Gaussian process underlying diffusion models.

### 4.1.1 ALIGNMENT IN THE LATENT-VARIABLE MANIFOLD

For a timestep $t$, let $\boldsymbol{\epsilon}_\theta(\boldsymbol{x}, t)$ denote the noise predicted from the DPMs. For a perturbation $\delta$ applied to the input latent variable $\boldsymbol{x}$, the first-order Taylor expansion of the corresponding change in the predicted noise satisfies

$$\Delta\boldsymbol{\epsilon} = \boldsymbol{\epsilon}_\theta(\boldsymbol{x} + \delta) - \boldsymbol{\epsilon}_\theta(\boldsymbol{x}) \approx \nabla_{\boldsymbol{x}}\boldsymbol{\epsilon}_\theta(\boldsymbol{x}, t)\delta. \tag{8}$$

From Equation (4), when the DPMs are well trained, the prediction of the the DPMs is the ideal scale score function, where $\boldsymbol{\epsilon}_\theta(x, t) \approx -\sigma_t \nabla_x \log q_t(x)$. Hence

$$\Delta\boldsymbol{\epsilon} \approx \nabla_{\boldsymbol{x}}\boldsymbol{\epsilon}_\theta(\boldsymbol{x}, t)\delta \approx -\sigma_t \nabla_x^2 \log q_t(\boldsymbol{x})\delta \tag{9}$$

**Gaussian regime.** When $t \to T$, the marginal distribution $q_t(\boldsymbol{x})$ approaches a Gaussian distribution:

$$q_t(\boldsymbol{x}) = \mathcal{N}(\boldsymbol{\mu}_t, \boldsymbol{\Sigma}_t) \approx \mathcal{N}(\mathbf{0}, \sigma_t^2 \mathbf{I}). \tag{10}$$

For a Gaussian distribution, the Hessian matrix of the log-density equals

$$H_t(x) = \nabla_{\boldsymbol{x}}^2 \log q_t(x) = -\boldsymbol{\Sigma}_t^{-1} = -(\sigma_t^2 \mathbf{I})^{-1}, \tag{11}$$

more details are shown in appendix.

Substituting this expression into Equation (9), the change in the predicted noise under a small input perturbation $\delta$ becomes

$$\Delta\boldsymbol{\epsilon} \approx \sigma_t \boldsymbol{\Sigma}_t^{-1} \delta = \frac{\sigma_t}{\sigma_t^2} \mathbf{I} \delta = \frac{1}{\sigma_t} \delta. \tag{12}$$

Thus, when the timestep $t$ is close to $T$, the latent variable distribution lies in the Gaussian regime. In this case, the change in the model's output is exactly aligned with the input perturbation. As a result, for any perturbation introduced to the input of DPMs, the corresponding predicted noise can be obtained without re-evaluating the DPM.

**No Gaussian regime.** When $t \to 0$, the marginal distribution $q_t(\boldsymbol{x})$ approaches the true data distribution. For a given forward process of DPMs $q(\boldsymbol{x}_t|\boldsymbol{x}_0) = \mathcal{N}(\boldsymbol{x}_t|\alpha_t\boldsymbol{x}_0, \sigma_t^2\mathbf{I})$, The latent variable $\boldsymbol{x}_t$ at timestep $t$ from a given latent variable $\boldsymbol{x}_{t-1}$ at timestep $t-1$ is calculated by

$$\boldsymbol{x}_t = \frac{\alpha_t}{\alpha_{t-1}}\boldsymbol{x}_{t-1} + \left(\sqrt{\sigma_t^2 - \frac{\alpha_t}{\alpha_{t-1}}\sigma_{t-1}^2}\right)\boldsymbol{\epsilon}, \tag{13}$$

where $\boldsymbol{\epsilon} \sim \mathcal{N}(\mathbf{0}, \mathbf{I})$. Let $\hat{\sigma}_t^2 = \sigma_t^2 - \frac{\alpha_t^2}{\alpha_{t-1}^2}\sigma_{t-1}^2$ and $\hat{\alpha}_t = \frac{\alpha_t}{\alpha_{t-1}}$

By using Tweedie's formula Efron (2011)

$$\nabla_{\boldsymbol{x}} \log q_t(\boldsymbol{x}) = \frac{\nabla_x q_t(\boldsymbol{x})}{q_t(\boldsymbol{x})} = \frac{\hat{\alpha}_t \mu_{t-1|t}(\boldsymbol{x}) - \boldsymbol{x}}{\hat{\sigma}_t^2}, \tag{14}$$

Where $\mu_{t-1|t} = \mathbb{E}(\boldsymbol{x}_{t-1}|\boldsymbol{x}_t) = \int \boldsymbol{x}_{t-1} p(\boldsymbol{x}_{t-1}|\boldsymbol{x}_t) d\boldsymbol{x}_{t-1}$.

The Hessian matrix is achieved by differentiating $\nabla_{\boldsymbol{x}} \log q_t(\boldsymbol{x})$, which is

$$\nabla_{\boldsymbol{x}}^2 \log q_t(\boldsymbol{x}) = \frac{\hat{\alpha}_t}{\hat{\sigma}_t^2}\nabla_{\boldsymbol{x}}\mu_{t-1|t}(\boldsymbol{x}) - \frac{1}{\hat{\sigma}_t^2}\mathbf{I} \tag{15}$$

$$= \frac{\hat{\alpha}_t^2}{\hat{\sigma}_t^4}\Sigma_{t-1|t}(\boldsymbol{x}) - \frac{1}{\hat{\sigma}_t^2}\mathbf{I},$$

we show the proof of $\nabla_{\boldsymbol{x}}\mu_{t-1|t}(\boldsymbol{x}) = \frac{\hat{\alpha}_t}{\hat{\sigma}^2}\Sigma_{t-1|t}(\boldsymbol{x})$ in appendix.

**Special case: treating the reverse transition as Gaussian.** When the timestep sizes are small, the reverse transitions are approximated by a Gaussian $q(\boldsymbol{x}_{t-1}|\boldsymbol{x}_t) = \mathcal{N}(\mu_{t-1|t}(\boldsymbol{x}_t), \Sigma_{t-1|t}(\boldsymbol{x}_t))$, where $\Sigma_{t-1|t}(\boldsymbol{x}) = \eta_t\mathbf{I}$ Sohl-Dickstein et al. (2015); Ho et al. (2020). Under this assumption, the Hessian matrix for each timestep is isotropic, where $\nabla_{\boldsymbol{x}}^2 \log q_t(\boldsymbol{x}) = \left(\frac{\hat{\alpha}_t^2}{\hat{\sigma}_t^4}\eta - \frac{1}{\hat{\sigma}_t^2}\right)\mathbf{I}$ indicate the corresponding predicted noise can be achieved by $\Delta_{\boldsymbol{\epsilon}} = \sigma_t\left(\frac{1}{\hat{\sigma}_t^2} - \frac{\hat{\alpha}_t^2}{\hat{\sigma}_t^4}\eta\right)\delta$

**General case.** However, as shown in Figure 1, the cosine similarity is a smooth decrease with the reverse process. This reveals that the reverse process of DPMs is close to, but not exactly, a Gaussian process, which indicates the $\Sigma_{t-1|t}(\boldsymbol{x})$ is not isotropic. Rewrite the Equation (15) to be $\nabla_{\boldsymbol{x}}^2 \log q_t(\boldsymbol{x}) = \frac{1}{\hat{\sigma}_t^2}\big(\frac{\hat{\alpha}_t^2}{\hat{\sigma}_t^2}\Sigma_{t-1|t}(\boldsymbol{x}) - \mathbf{I}\big)$, by using Laplace approximation Bishop (2006).

$$\nabla_{\boldsymbol{x}}^2 \log q_t(\boldsymbol{x}) \approx \frac{\hat{\alpha}_t^2}{\hat{\sigma}_t^4}\big(-Hm(\boldsymbol{x}_{t-1|t}) + \frac{\hat{\alpha}_t^2}{\hat{\sigma}_t^2}\mathbf{I}\big)^{-1} - \frac{1}{\hat{\sigma}_t^2}\mathbf{I}, \tag{16}$$

where $m(\boldsymbol{x})$ is the maximum a posteriori (MAP) estimate of $\boldsymbol{x}_{t-1}$ for a given $\boldsymbol{x}_t$. As the sampling methods reverse toward the data distribution, the geometry of $\boldsymbol{x}$ reflects the natural anisotropy of the data. Hence, isotropy is progressively lost as $t$ approaches zero. For example, the isotropy of Stable diffusion is decreasing as shown in Figure 1.

Since the isotropy of the Hessian matrix varies throughout the sampling process, as shown in Figure 1, the reverse process maintains a high degree of isotropy for the majority of its duration. This property can therefore be exploited as a "free lunch" adjustment to refine latent variables, integrating seamlessly with existing sampling methods. Importantly, this plug-in refinement incurs no additional computational cost. In the following sections, we demonstrate how this characteristic can be applied to two types of sampling methods: sequential sampling methods and parallel sampling methods.

## 4.2 ALLEVIATING THE ERROR IN SEQUENTIAL SAMPLING METHODS

Score-based MCMC approaches have been proposed to address the reverse process of DPMs within the predictor-corrector framework Song et al. (2020); Zhao et al. (2023). This framework consists of two components: Predictor and Corrector. Predictor estimates the latent variable at the next time step, and corrector is used to mitigate the discretization error introduced during prediction. Together, these components enhance the accuracy and efficiency of the sampling process.

By integrating our proposed method into this framework, we further improve sampling efficiency by reducing the discrepancy between the refined latent variable and the predicted noise from the unrefined latent variable. This alignment leads to more consistent updates across iterations, improved sample quality.

The corrector refines the latent variable at timestep $t$ via Equation (7) However, in the standard predictor-corrector framework, the subsequent predictor still uses the noise predicted at the unrefined latent variable, $\boldsymbol{x}_{t-1} \leftarrow \text{Predictor}\big(\boldsymbol{x}_t^c, \boldsymbol{\epsilon}_\theta(\boldsymbol{x}_t, t), Q\big)$. Rather than re-evaluating the model at the corrected variable. For comparison, a recomputed variant would use $\boldsymbol{x}_{t-1} \leftarrow \text{Predictor}\big(\boldsymbol{x}_t^c, \boldsymbol{\epsilon}_\theta(\boldsymbol{x}_t^c, t), Q\big)$, where the noise is updated by the refined latent variable $\boldsymbol{x}_t^c$. The difference between $\boldsymbol{\epsilon}(\boldsymbol{x}_t, t)$ and $\boldsymbol{\epsilon}(\boldsymbol{x}_t^c, t)$ introduces additional error into the sampling process, particularly when the corrector significantly adjusts the latent variable. This inconsistency may undermine the benefits of the corrector and reduce overall sampling fidelity.

Motivated by the geometric properties revealed in our analysis, we propose a method that leverages the difference between the refined and unrefined latent variables to adjust the predicted noise. This adjustment effectively alleviates the additional error introduced by the mismatch between the latent variables used in the corrector and the predictor steps without any extra computational cost or requiring changes to the training process. Specifically, we modify the predictor step as follows:

$$\boldsymbol{x}_{t-1} \leftarrow \text{Predictor}\big(\boldsymbol{x}_t^c, \boldsymbol{\epsilon}_\theta(\boldsymbol{x}_t, t) + \lambda_t\big(\boldsymbol{x}_t^c - \boldsymbol{x}_t\big), Q\big), \tag{17}$$

where $\lambda_t$ is a coefficient that controls the correction term and $\boldsymbol{\epsilon}_t^c = \boldsymbol{\epsilon}_\theta(\boldsymbol{x}_t, t) + \lambda_t\big(\boldsymbol{x}_t^c - \boldsymbol{x}_t\big)$ in Figure 2. The pseudo code is shown in Algorithm 1.

## 4.3 ACCELERATING THE CONVERGENCE OF PARALLEL SAMPLING

Conventional sampling methods for DPMs process through all timesteps sequentially, which limits hardware utilization. To better leverage multiple GPUs, parallel sampling methods are proposed by Shih et al. (2023) that reformulate sequential sampling by applying Picard iteration so that many timesteps can be processed in parallel. However, such parallel sampling typically requires additional evaluations of the DPMs to reach convergence. We accelerate this approach with a dual-update for each model call, improving convergence speed and reducing the total number of evaluations.

---

**Algorithm 1** Free Lunch for Sequential Methods

---

**Input**: A diffusion model $\epsilon_\theta(\boldsymbol{x}_t, t)$, an initial random noise $\boldsymbol{x}_T \sim \mathcal{N}(\boldsymbol{0}, \mathbf{I})$, Total timesteps of DPMs $T$, any solver $\mathbf{P}$ for predictor and $\mathbf{C}$ for corrector, query $Q = \{\epsilon_i\}$ for containing previous predicted noise.

1: $Q \leftarrow \{\epsilon_\theta(\boldsymbol{x}_T, T)\}$
2: $t \leftarrow T - 1$
3: $\boldsymbol{x}_{t+1} \leftarrow \boldsymbol{x}_T$
4: **while** $t > 0$ **do**
5:     $\boldsymbol{x}_t \leftarrow \mathbf{P}(\boldsymbol{x}_{t+1}, Q)$          $\triangleright$ Use predictor to get the predicted latent variable at timestep $t$.
6:     $\epsilon_t \leftarrow \epsilon_\theta(\boldsymbol{x}_t, t)$
7:     $Q \leftarrow \{\epsilon_t, Q\}$          $\triangleright$ Append the predicted noise to the $Q$
8:     $\boldsymbol{x}_t^c \leftarrow \mathbf{C}(\boldsymbol{x}_{t+1}, Q)$      $\triangleright$ Use corrector to correct predicted latent variable at timestep $t$.
9:     $Q \leftarrow \{\epsilon_t + \lambda_t(\boldsymbol{x}_t^c - \boldsymbol{x}_t), Q \backslash \{\epsilon_t\}\}$      $\triangleright$ Update the predicted noise at timestep $t$ in $Q$
10: **end while**

---

Following Shih et al. (2023), one Picard iteration of parallel sampling updates

$$\boldsymbol{x}_t^{k+1} = \boldsymbol{x}_T^k + \sum_{i=T}^{t+1} \text{Predictor}(\boldsymbol{x}_i^k, \epsilon_\theta(\boldsymbol{x}_i^k, i), Q) - \boldsymbol{x}_i^k, \tag{18}$$

where $t = 0, \dots, T - 1$.

**Dual-update.** We perform a second update that reuses the latent variables predicted at iteration $k + 1$:

$$\boldsymbol{x}_t^{k+2} = \boldsymbol{x}_T^{k+1} + \sum_{i=T}^{t+1} \text{Predictor}(\boldsymbol{x}_i^{k+1}, \epsilon_\theta(\boldsymbol{x}_i^k, i) + \lambda_i(\boldsymbol{x}_t^{k+1} - \boldsymbol{x}_i^k), Q) - \boldsymbol{x}_i^{k+1}. \tag{19}$$

We refer to this as a dual-update because it performs two iterations of updates per DPMs evaluation. In the appendix, we explore multi-update cases and show that after the dual-update, the displacement between iterations is small. Pseudocode for our parallel sampling method based on ParaDiGMS Shih et al. (2023) is provided in Algorithm 2.

---

**Algorithm 2** Free Lunch for Parallel Methods by Dual-Update

---

**Input**: Diffusion model $\epsilon_\theta(\boldsymbol{x}_t, t)$, initial random noise $\boldsymbol{x}_T \sim \mathcal{N}(\boldsymbol{0}, \mathbf{I})$, Total time steps of DPMs is $T$, any solver $\mathbf{P}$ for predictor, tolerance $\tau$, window size $p$.

1: $t \leftarrow T$
2: $k \leftarrow 0$
3: $\boldsymbol{x}_{t-i}^k \leftarrow \boldsymbol{x}_T \; \forall i \in [0, p)$
4: **while** $t > 0$ **do**
5:     $\epsilon_{t-i}^k \leftarrow \epsilon_\theta(\boldsymbol{x}_{t-i}^k, t-i) \quad \forall i \in [0, p)$      $\triangleright$ Get the predicted noise in parallel.
6:     $\boldsymbol{y}_{t-i} \leftarrow \mathbf{P}(x_{t-i}^k, \epsilon_{t-i}^k) - x_{t-i}^k \quad \forall i \in [0, p)$      $\triangleright$ Compute drifts in parallel.
7:     $\boldsymbol{x}_{t-i-1}^{k+1} \leftarrow \boldsymbol{x}_t^k + \sum_{j=t}^{t-i} \boldsymbol{y}_j \quad \forall i \in [0, p)$      $\triangleright$ Picard iteration.
8:     $\tilde{\boldsymbol{y}}_{t-i} \leftarrow \mathbf{P}(\boldsymbol{x}_{t-i}^{k+1}, \epsilon_{t-i}^k + \lambda_{t-i}(\boldsymbol{x}_{t-i}^{k+1} - \boldsymbol{x}_{t-i}^k)) - \boldsymbol{x}_{t-i}^{k+1} \quad \forall i \in [0, p)$   $\triangleright$ Dual-update.
9:     $\boldsymbol{x}_{t-i-1}^{k+2} \leftarrow \boldsymbol{x}_t^{k+1} + \sum_{j=t}^{t-i} \tilde{\boldsymbol{y}}_j \quad \forall i \in [0, p)$
10:    $\text{error}_i \leftarrow \frac{1}{D} \|x_{t-i}^{k+2} - x_{t-i}^k\|_2^2 \quad \text{for all } i \in [1, p)$
11:    $\text{stride} \leftarrow \min\left(\{i : \text{error}_i > \tau^2 \sigma_{t-i}^2\} \cup \{p\}\right)$      $\triangleright$ Slide window until tolerance.
12:    $\boldsymbol{x}_{t-i}^{k+2} \leftarrow \boldsymbol{x}_{t-i}^{k+2} \quad \text{for all } i \in [1, \text{stride}]$      $\triangleright$ Start new coverage.
13:    $t \leftarrow t - \text{stride}; \; k \leftarrow k + 2; \; p \leftarrow \min(p, T - t)$
14: **end while**

---

## 5 EXPERIMENTS

In this section, we integrate our method with several state-of-the-art (SOTA) sampling algorithms to evaluate its effectiveness and efficiency. Notably, our method requires neither preparation overhead nor additional evaluations of DPMs. For sequential sampling methods, our technique enhances image quality as a plug-in refinement. For parallel sampling methods, we propose dual-update to accelerate convergence without degrading sample quality. We use $-Hm(\boldsymbol{x}_{t-1|t}) = 1/\sigma_{t-1}^2$ in Equation (16) as the default setting of $\lambda$ in Algorithm 1 and Algorithm 2.

We evaluate our method on CIFAR-10 Krizhevsky et al. (2009), ImageNet $256 \times 256$ Deng et al. (2009), and LSUN Bedroom $256 \times 256$ Yu et al. (2015). Our baselines include UniPC Zhao et al. (2023), DPM-Solver-v3 Zheng et al. (2023), AMED-Plugin Zhou et al. (2024), and ParaDiGMS Shih et al. (2023).

### 5.1 SEQUENTIAL SAMPLE METHOD

We first evaluate our method in the sequential sampling method. We deploy it as a plug-in to existing solvers neither preparation overhead nor additional evaluations (*e.g.* UniPC Zhao et al. (2023) and DPM-Solver-v3 Zheng et al. (2023)). On ImageNet $256 \times 256$ and LSUN Bedroom $256 \times 256$, it consistently improves the quality of generated images at a fixed number of function evaluations (NFE). Moreover, the execution time of image generation remains unchanged after integrating our method.

Table 1 shows the results on LSUN bedroom $256 \times 256$. Following the setting in DPM-Solver-v3 Zheng et al. (2023), we use a stable diffusion model to generate 50k images for evaluation. Table 2 reports results on ImageNet $256 \times 256$. Using a guided diffusion model, we generate 50k images for evaluation. Across both benchmarks, our method consistently improves the image quality over SOTA baselines. In tables, we refer to DPM-Solver-v3 as DPM-v3. Notably, our method is **zero-cost** and operates as a **test-time** plug-in. Additional experiments are provided in the Appendix.

Table 1: Experiments on LSUN bedroom $256 \times 256$ guided with various NFE. The quality of images is measured by FID. We use the implementation from the DPM-v3 repository.

| NFE | 3 | 4 | 5 |
|---|---|---|---|
| UniPC | 109.31 | 39.89 | 13.99 |
| + Ours | 59.27 | 20.08 | 8.96 |
| Δ | (-50.04) | (-19.81) | (-5.03) |
| DPM-v3 | 64.43 | 19.17 | 8.96 |
| + Ours | 54.41 | 15.49 | 6.77 |
| Δ | (-10.02) | (-3.68) | (-2.19) |

Table 2: Experiments on ImageNet $256 \times 256$ guided with various NFE. The quality of images is measured by FID. We use the implementation from the DPM-v3 repository.

| NFE | 3 | 4 | 5 |
|---|---|---|---|
| UniPC | 52.21 | 24.53 | 15.62 |
| + Ours | 50.79 | 22.60 | 14.38 |
| Δ | (-1.42) | (-1.93) | (-1.24) |
| DPM-v3 | 65.38 | 26.37 | 15.10 |
| + Ours | 60.80 | 24.34 | 14.53 |
| Δ | (-4.58) | (-2.03) | (-0.57) |

Figure 3 shows generated images using the UniPC and DPM-Solver-v3, both in its vanilla form and combined with our refinement method. As illustrated, our approach produces samples with sharper details, while maintaining the overall style and diversity of the baseline solver. These qualitative results complement our quantitative evaluation, confirming that integrating our method into existing solvers improves visual quality without additional computational cost.

### 5.2 PARALLEL SAMPLE METHOD

For parallel sampling based on Picard iteration, we introduce a dual-update mechanism within each iteration to accelerate convergence. Specifically, we integrate our method into the ParaDiGMS framework Shih et al. (2023). After the standard Picard update, we apply an additional refinement step that yields extra improvement per DPMs evaluation. As demonstrated in Table 3, this dual-update strategy consistently reduces both the number of iterations and the overall sampling time, while maintaining comparable sample quality.

Following ParaDiGMS Shih et al. (2023), we compute CLIP scores using ViT-b-14 Dosovitskiy et al. (2021) from the Hugging Face implementation (checkpoint openai/clip-vit-base-patch16). In the original ParaDiGMS setup, the stopping tolerance is set to 0.1. For a fair comparison, we adjust the tolerance in our method to achieve a comparable CLIP score, and report the corresponding iteration counts and times needed for each sample. As shown in Table 3, integrating our method reduces computational cost and sampling time by over 10% while achieving results comparable to the original ParaDiGMS.

Table 3: Experiments on COCO Lin et al. (2014). Using Stable Diffusion v2, we generate images conditioned on the 1,000 captions from the COCO 2017 annotations. Image quality is measured by the CLIP score, computed with the implementation provided in the ParaDiGMS repository. We use ParaDiGMS (500) indicates 500 timesteps.

|  | Window Size | Model Evals | Parallel Iters | Clip Score | Time/Sample |
|---|---|---|---|---|---|
| ParaDiGMS (500) | 5 | 561 | 113 | 31.78 | 20.93s |
| + Ours | 5 | 507 | 102 | 31.78 | 18.78s |
| ParaDiGMS (500) | 10 | 652 | 66 | 31.76 | 23.47s |
| + Ours | 10 | 548 | 55 | 31.75 | 19.51s |
| ParaDiGMS (200) | 5 | 243 | 49 | 31.75 | 9.17s |
| + Ours | 5 | 217 | 44 | 31.75 | 7.84s |
| ParaDiGMS (200) | 10 | 286 | 30 | 31.74 | 10.41s |
| + Ours | 10 | 250 | 26 | 31.73 | 8.58s |

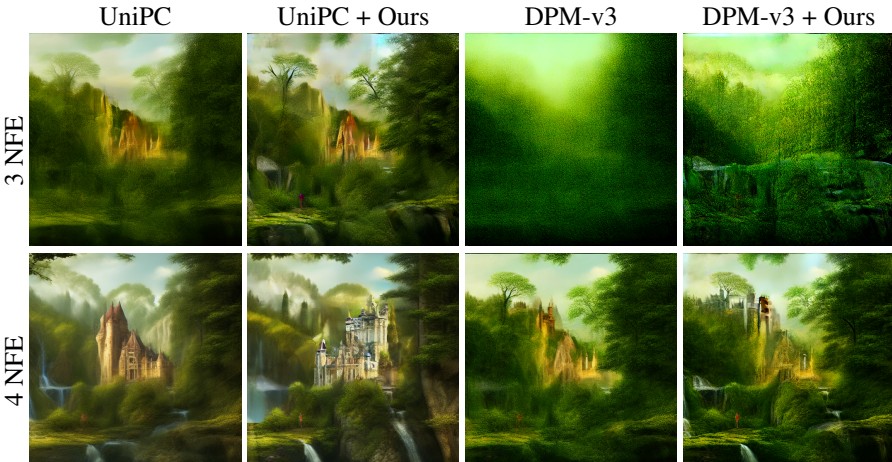

Figure 3: Visualization of generated images using Stable-Diffusion-v1.4 of UniPC, DPM-Solver-v3, and our method integrated with 3 and 4 NFEs. Prompt: "A beautiful castle beside a waterfall in the woods, by Josef Thoma, matte painting, trending on ArtStation HQ.". We refer to DPM-Solver-v3 as DPM-v3.

## 6 CONCLUSION

In this work, we shift the focus from solver design to the underlying geometry of DPMs. By uncovering and formalizing an isotropic property induced by the Gaussian process of DPMs, we provide a new perspective on the latent manifold of DPMs. Building on this insight, we propose a lightweight test-time refinement method that integrates seamlessly into existing sampling frameworks, reducing discretization error and improving stability without extra training or computational overhead. Moreover, we demonstrate that our approach accelerates parallel sampling methods such as Picard iteration, yielding consistent gains in both efficiency and fidelity. We hope that this work encourages further research into the geometric foundations of DPMs. A deeper understanding of these structures may lead to new algorithms that better balance sample quality with generation cost.

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

# Appendix

## Table of Contents

## A  ACKNOWLEDGMENT OF LLM USAGE

We used a large language model (ChatGPT) to polish this paper. Its use was limited to grammar checking, fixing typos, rephrasing sentences for clarity, and improving word choice. All conceptual contributions, methodological designs, experiments, and analyses were carried out entirely by the authors. The use of an LLM does not affect the reproducibility or scientific validity of our work.

## B  PROOF

**Proof of Equation (11)**   . For a Gaussian distribution $q(\boldsymbol{x}) \approx \mathcal{N}(\boldsymbol{\mu}, \boldsymbol{\Sigma})$, where $\boldsymbol{\mu} \in \mathbb{R}^n$ and $\boldsymbol{\Sigma}_t \in \mathbb{R}^{n \times n}$.

$$\log q(\boldsymbol{x}) = \log\left((2\pi)^{\frac{-n}{2}}\right) + \log\left(\det(\boldsymbol{\Sigma}_t)^{-\frac{1}{2}}\right) - \frac{1}{2}(\boldsymbol{x} - \boldsymbol{\mu})^\top \boldsymbol{\Sigma}_t^{-1}(\boldsymbol{x} - \boldsymbol{\mu}). \tag{20}$$

The Jacobian matrix and Hessian matrix are

$$\nabla \log q(\boldsymbol{x}) = -\boldsymbol{\Sigma}_t^{-1}(\boldsymbol{x} - \boldsymbol{\mu}), \nabla^2 \log q(\boldsymbol{x}) = \boldsymbol{\Sigma}_t^{-1}. \tag{21}$$

**Proof of Equation (15)**   As $\mu_{t-1|t}(\boldsymbol{x}) = \mathbb{E}(\boldsymbol{x}_{t-1}|\boldsymbol{x}_t) = \int \boldsymbol{x}_{t-1} q(\boldsymbol{x}_{t-1}|\boldsymbol{x}_t) d\boldsymbol{x}_{t-1}$, the derivative of $\mu_{t-1|t}$ is

$$\nabla_{\boldsymbol{x}_t} \mu_{t-1|t}(\boldsymbol{x}_t) = \nabla_{\boldsymbol{x}_t} \int \boldsymbol{x}_{t-1} q(\boldsymbol{x}_{t-1}|\boldsymbol{x}_t) d\boldsymbol{x}_{t-1} = \int \boldsymbol{x}_{t-1} \nabla_{\boldsymbol{x}_t} q(\boldsymbol{x}_{t-1}|\boldsymbol{x}_t) d\boldsymbol{x}_{t-1} \tag{22}$$

For $\nabla_{\boldsymbol{x}_t} q(\boldsymbol{x}_{t-1}|\boldsymbol{x}_t)$ part, using bayes rule to get

$$\nabla_{\boldsymbol{x}_t} q(\boldsymbol{x}_{t-1}|\boldsymbol{x}_t) = \nabla_{\boldsymbol{x}_t}\left(\frac{q(\boldsymbol{x}_t|\boldsymbol{x}_{t-1})q(\boldsymbol{x}_{t-1})}{q(\boldsymbol{x}_t)}\right) \tag{23}$$

$$= q(\boldsymbol{x}_{t-1})\nabla_{\boldsymbol{x}_t}\left(\frac{q(\boldsymbol{x}_t|\boldsymbol{x}_{t-1})}{q(\boldsymbol{x}_t)}\right)$$

$$= q(\boldsymbol{x}_{t-1})\left(\frac{\nabla_{\boldsymbol{x}_t} q(\boldsymbol{x}_t|\boldsymbol{x}_{t-1})}{q(\boldsymbol{x}_t)} - \frac{q(\boldsymbol{x}_t|\boldsymbol{x}_{t-1})\nabla_{\boldsymbol{x}_t} q(\boldsymbol{x}_t)}{q(\boldsymbol{x}_t)^2}\right)$$

$$= \frac{q(\boldsymbol{x}_{t-1})q(\boldsymbol{x}_t|\boldsymbol{x}_{t-1})}{q(\boldsymbol{x}_t)}\left(\frac{\nabla_{\boldsymbol{x}_t} q(\boldsymbol{x}_t|\boldsymbol{x}_{t-1})}{q(\boldsymbol{x}_t|\boldsymbol{x}_{t-1})} - \frac{\nabla_{\boldsymbol{x}_t} q(\boldsymbol{x}_t)}{q(\boldsymbol{x}_t)}\right)$$

$$= q(\boldsymbol{x}_{t-1}|\boldsymbol{x}_t)\left(\nabla_{\boldsymbol{x}_t} \log q(\boldsymbol{x}_t|\boldsymbol{x}_{t-1}) - \nabla_{\boldsymbol{x}_t} \log q(\boldsymbol{x}_t)\right)$$

Substitude $\nabla_{\boldsymbol{x}_t} q(\boldsymbol{x}_{t-1}|\boldsymbol{x}_t)$ in Equation (22)

$$\nabla_{\boldsymbol{x}_t} \mu_{t-1|t}(\boldsymbol{x}_t) = \int \boldsymbol{x}_{t-1} q(\boldsymbol{x}_{t-1}|\boldsymbol{x}_t) \nabla_{\boldsymbol{x}_t} \log q(\boldsymbol{x}_t|\boldsymbol{x}_{t-1}) d\boldsymbol{x}_{t-1} \tag{24}$$

$$- \int \boldsymbol{x}_{t-1} q(\boldsymbol{x}_{t-1}|\boldsymbol{x}_t) \nabla_{\boldsymbol{x}_t} \log q(\boldsymbol{x}_t) d\boldsymbol{x}_{t-1}$$

As forward diffusion process is Gaussian process $q(\boldsymbol{x}_t; \boldsymbol{x}_{t-1}) = \mathcal{N}(\boldsymbol{x}_t|\hat{\alpha}_t \boldsymbol{x}_{t-1}, \hat{\sigma}_t^2 I)$, the partial derivative of $\log q(\boldsymbol{x}_t|\boldsymbol{x}_{t-1})$ with respect to latent variable $\boldsymbol{x}_t$ is

$$\nabla_{\boldsymbol{x}_t} \log q(\boldsymbol{x}_t|\boldsymbol{x}_{t-1}) = -\frac{\boldsymbol{x}_t - \hat{\alpha}_t \boldsymbol{x}_{t-1}}{\hat{\sigma}_t^2}. \tag{25}$$

The first term of Equation (24) can be calculated by

$$\int \boldsymbol{x}_{t-1} q(\boldsymbol{x}_{t-1}|\boldsymbol{x}_t) \nabla_{\boldsymbol{x}} \log q(\boldsymbol{x}_t|\boldsymbol{x}_{t-1}) d\boldsymbol{x}_{t-1} = \int \boldsymbol{x}_{t-1} \left( \frac{\hat{\alpha}_t \boldsymbol{x}_{t-1} - \boldsymbol{x}_t}{\hat{\sigma}_t^2} \right) q(\boldsymbol{x}_{t-1}|\boldsymbol{x}_t) d\boldsymbol{x}_{t-1} \tag{26}$$

$$= \frac{\hat{\alpha}_t}{\hat{\sigma}_t^2} \int \boldsymbol{x}_{t-1} \left( \boldsymbol{x}_{t-1} - \frac{\boldsymbol{x}_t}{\hat{\alpha}_t} \right) q(\boldsymbol{x}_{t-1}|\boldsymbol{x}_t) d\boldsymbol{x}_{t-1}$$

Decompose the term $\boldsymbol{x}_{t-1}(\hat{\alpha}_t \boldsymbol{x}_{t-1} - \boldsymbol{x}_t)$ in Equation (26) by

$$\boldsymbol{x}_{t-1} \left( \boldsymbol{x}_{t-1} - \frac{\boldsymbol{x}_t}{\hat{\alpha}_t} \right) \tag{27}$$

$$= (\boldsymbol{x}_{t-1} - \overline{\boldsymbol{x}}_{t-1} + \overline{\boldsymbol{x}}_{t-1})(\boldsymbol{x}_{t-1} - \overline{\boldsymbol{x}}_{t-1} + \overline{\boldsymbol{x}}_{t-1} - \frac{\boldsymbol{x}_t}{\hat{\alpha}_t})$$

$$= (\boldsymbol{x}_{t-1} - \overline{\boldsymbol{x}}_{t-1})(\boldsymbol{x}_{t-1} - \overline{\boldsymbol{x}}_{t-1}) + \overline{\boldsymbol{x}}_{t-1}(\boldsymbol{x}_{t-1} - \overline{\boldsymbol{x}}_{t-1})$$

$$+ (\boldsymbol{x}_{t-1} - \overline{\boldsymbol{x}}_{t-1})(\overline{\boldsymbol{x}}_{t-1} - \frac{\boldsymbol{x}_t}{\hat{\alpha}_t}) + \overline{\boldsymbol{x}}_{t-1}(\overline{\boldsymbol{x}}_{t-1} - \frac{\boldsymbol{x}_t}{\hat{\alpha}_t}).$$

Integrating term by term gives

$$\int (\boldsymbol{x}_{t-1} - \overline{\boldsymbol{x}}_{t-1})(\boldsymbol{x}_{t-1} - \overline{\boldsymbol{x}}_{t-1}) q(\boldsymbol{x}_{t-1}|\boldsymbol{x}_t) d\boldsymbol{x}_{t-1} = \Sigma_{t-1|t}(\boldsymbol{x}), \tag{28}$$

$$\int \overline{\boldsymbol{x}}_{t-1}(\boldsymbol{x}_{t-1} - \overline{\boldsymbol{x}}_{t-1}) q(\boldsymbol{x}_{t-1}|\boldsymbol{x}_t) d\boldsymbol{x}_{t-1} = 0, \tag{29}$$

$$\int (\boldsymbol{x}_{t-1} - \overline{\boldsymbol{x}}_{t-1})(\overline{\boldsymbol{x}}_{t-1} - \boldsymbol{x}_t) q(\boldsymbol{x}_{t-1}|\boldsymbol{x}_t) d\boldsymbol{x}_{t-1} = 0, \tag{30}$$

$$\int \overline{\boldsymbol{x}}_{t-1}(\overline{\boldsymbol{x}}_{t-1} - \frac{\boldsymbol{x}_t}{\hat{\alpha}_t}) q(\boldsymbol{x}_{t-1}|\boldsymbol{x}_t) d\boldsymbol{x}_{t-1} = \overline{\boldsymbol{x}}_{t-1}(\overline{\boldsymbol{x}}_{t-1} - \frac{\boldsymbol{x}_t}{\hat{\alpha}_t}). \tag{31}$$

Therefore, the Equation (26), which is the first term of Equation (24):

$$\int \boldsymbol{x}_{t-1} q(\boldsymbol{x}_{t-1}|\boldsymbol{x}_t) \nabla_{\boldsymbol{x}} \log q(\boldsymbol{x}_t|\boldsymbol{x}_{t-1}) d\boldsymbol{x}_{t-1} = \frac{\hat{\alpha}_t}{\hat{\sigma}_t^2} \left( \Sigma_{t-1|t}(\boldsymbol{x}) + \overline{\boldsymbol{x}}_{t-1}(\overline{\boldsymbol{x}}_{t-1} - \frac{\boldsymbol{x}_t}{\hat{\alpha}_t}) \right). \tag{32}$$

The second term of Equation (24) is calculated by

$$\int \boldsymbol{x}_{t-1} q(\boldsymbol{x}_{t-1}|\boldsymbol{x}_t) \nabla_{\boldsymbol{x}_t} \log q(\boldsymbol{x}_t) d\boldsymbol{x}_{t-1} = \overline{\boldsymbol{x}}_{t-1} \nabla_{\boldsymbol{x}_t} \log q(\boldsymbol{x}_t). \tag{33}$$

Decompose the term $\nabla_{\boldsymbol{x}_t} \log q(\boldsymbol{x}_t)$ by

$$\nabla_{\boldsymbol{x}} \log q(\boldsymbol{x}_t) = \frac{\nabla_{\boldsymbol{x}_t} q(\boldsymbol{x}_t)}{q(\boldsymbol{x}_t)} \tag{34}$$

$$= \frac{\int \nabla_{\boldsymbol{x}_t} q(\boldsymbol{x}_t|\boldsymbol{x}_{t-1})q(\boldsymbol{x}_{t-1})d\boldsymbol{x}_{t-1}}{q(\boldsymbol{x}_t)}$$

$$= \frac{\int q(\boldsymbol{x}_t|\boldsymbol{x}_{t-1})\nabla_{\boldsymbol{x}_t} \log q(\boldsymbol{x}_t|\boldsymbol{x}_{t-1})q(\boldsymbol{x}_{t-1})d\boldsymbol{x}_{t-1}}{q(\boldsymbol{x}_t)}$$

$$= \frac{\int q(\boldsymbol{x}_t|\boldsymbol{x}_{t-1})\frac{\hat{\alpha}_t}{\hat{\sigma}_t^2}(\boldsymbol{x}_{t-1} - \frac{\boldsymbol{x}_t}{\hat{\alpha}_t})q(\boldsymbol{x}_{t-1})d\boldsymbol{x}_{t-1}}{q(\boldsymbol{x}_t)}$$

$$= \int \frac{q(\boldsymbol{x}_t|\boldsymbol{x}_{t-1})q(\boldsymbol{x}_{t-1})}{q(\boldsymbol{x}_t)}\frac{\hat{\alpha}_t}{\hat{\sigma}_t^2}(\boldsymbol{x}_{t-1} - \frac{\boldsymbol{x}_t}{\hat{\alpha}_t})d\boldsymbol{x}_{t-1}$$

$$= \frac{\hat{\alpha}_t}{\hat{\sigma}_t^2}\int q(\boldsymbol{x}_{t-1}|\boldsymbol{x}_t)(\boldsymbol{x}_{t-1} - \frac{\boldsymbol{x}_t}{\hat{\alpha}_t})d\boldsymbol{x}_{t-1}$$

$$= \frac{\hat{\alpha}_t}{\hat{\sigma}_t^2}(\overline{\boldsymbol{x}}_{t-1} - \frac{\boldsymbol{x}_t}{\hat{\alpha}_t})$$

Therefore, the second term of Equation (24)

$$\int \boldsymbol{x}_{t-1}q(\boldsymbol{x}_{t-1}|\boldsymbol{x}_t)\nabla_{\boldsymbol{x}} \log q(\boldsymbol{x}_t)d\boldsymbol{x}_{t-1} = \frac{\hat{\alpha}_t}{\hat{\sigma}_t^2}\overline{\boldsymbol{x}}_{t-1}(\overline{\boldsymbol{x}}_{t-1} - \frac{\boldsymbol{x}_t}{\hat{\alpha}_t}) \tag{35}$$

Substitute the first term and second term in Equation (24) by Equation (32) and Equation (35) to achieve

$$\nabla_{\boldsymbol{x}}\mu_{t-1|t}(\boldsymbol{x}) = \frac{\hat{\alpha}_t}{\hat{\sigma}_t^2}\Sigma_{t-1|t}(\boldsymbol{x}) \tag{36}$$

Therefore, the Hessian matrix in Equation (15) is

$$\nabla_{\boldsymbol{x}}^2 \log q_t(\boldsymbol{x}) = \frac{\hat{\alpha}_t^2}{\hat{\sigma}_t^4}\Sigma_{t-1|t}(\boldsymbol{x}) - \frac{1}{\hat{\sigma}_t^2}\mathbf{I} \tag{37}$$

## C  MORE EXPERIMENTS

Table 4 and Table 5 report results on LSUN bedroom 256×256 and ImageNet 256×256 across NEF from 3 to 20. Our method yields consistent gains at every NEF, improving FID over all baselines without extra calculation.

Table 4: Experiments on LSUN bedroom 256×256 guided with various NFE. The quality of images is measured by FID. We use the implementation from the DPM-Solver-v3 repository. * We borrow results reported in AMED-Plugin, "(NEF)" denotes the actual NEF corresponding to the reported FID.

| NFE | 3 | 4 | 5 | 6 | 8 | 10 | 12 | 15 | 20 |
|---|---|---|---|---|---|---|---|---|---|
| AMED-Plugin* | 101.5 | - | 25.68 | 8.63 (7) | 7.82 (9) | - | - | - | - |
| UniPC | 109.31 | 39.89 | 13.99 | 6.55 | 4.00 | 3.57 | 3.35 | 3.18 | 3.07 |
| + Ours | 59.27 | 20.08 | 8.96 | 5.48 | 3.79 | 3.38 | 3.19 | 3.08 | 3.02 |
| DPM-Solver-v3 | 64.43 | 19.17 | 8.96 | 5.13 | 3.56 | 3.20 | 3.12 | 3.12 | 3.10 |
| + Ours | 54.41 | 15.49 | 6.77 | 4.55 | 3.45 | 3.14 | 3.05 | 3.04 | 3.04 |

## D  MULTI UPDATE

In Table 6, we examine the effect of applying multiple updates within ParaDiGMS. In particular, we compare the latent variable before each Picard iteration with the updated latent variable after the iteration when our refinement method is applied.

Table 5: Experiments on ImageNet 256×256 guided with various NFE. The quality of images is measured by FID. We use the implementation from the DPM-Solver-v3 repository. * We borrow the results (NFE∈[5,20]) reported in DPM-Solver-v3.

| NFE | 3 | 4 | 5 | 6 | 8 | 10 | 12 | 15 | 20 |
|---|---|---|---|---|---|---|---|---|---|
| UniPC* | 52.21 | 24.53 | 15.62 | 11.91 | 9.29 | 8.35 | 7.95 | 7.64 | 7.44 |
| + Ours | 50.79 | 22.60 | 14.38 | 11.07 | 9.06 | 8.26 | 7.97 | 7.74 | 7.46 |
| DPM-Solver-v3* | 65.38 | 26.37 | 15.10 | 11.39 | 8.96 | 8.27 | 7.94 | 7.62 | 7.39 |
| + Ours | 60.80 | 24.34 | 14.53 | 10.84 | 8.86 | 8.08 | 7.85 | 7.55 | 7.46 |

Table 6: Experiments on multi-update

| Number of Iters | 1 | 2 | 3 | 4 | 5 |
|---|---|---|---|---|---|
| Difference | 51.7352 | 1.2923 | 1.2919 | 1.2919 | 1.2919 |

## E ABLATION STUDY

Table 7 presents an ablation study on the choice of the parameter $\lambda$. We compare different strategies, including the default setting $-Hm(\boldsymbol{x}_{t-1|t}) = \frac{1}{\sigma_{t-1}^2}$, fixed constant values, and empirical values collected from experiments.

Table 7: Experiments on LSUN bedroom 256×256 guided with various NFE. The quality of images is measured by FID. We use the implementation from the DPM-Solver-v3 repository.

| NFE | 3 | 4 | 5 | 6 | 8 | 10 | 12 | 15 | 20 |
|---|---|---|---|---|---|---|---|---|---|
| DPM-Solver-v3 | 64.43 | 19.17 | 8.96 | 5.13 | 3.56 | 3.20 | 3.12 | 3.12 | 3.10 |
| + Ours | 54.41 | 15.49 | 6.77 | 4.55 | 3.45 | 3.14 | 3.05 | 3.04 | 3.04 |
| + Ours $\lambda = 1$ | 63.42 | 18.71 | 7.42 | 4.76 | 3.52 | 3.18 | 3.09 | 3.08 | 3.06 |
| + Ours $\lambda = 0.5$ | 49.44 | 14.68 | 6.74 | 4.37 | 3.42 | 3.14 | 3.05 | 3.05 | 3.05 |
| + Ours empirical | 53.17 | 15.61 | 6.89 | 4.61 | 3.48 | 3.16 | 3.07 | 3.06 | 3.06 |

## F CHECKPOINTS

For all experiments, we employ publicly released model checkpoints to ensure reproducibility and fair comparison. In particular, we reference the official implementations of DPM-Solver-v3 and ParaDiGMS, and list the specific checkpoints used in Table 8.

## G MORE VISUALIZATION

Figure 4 presents additional visualizations of generated images on the LSUN Bedroom 256×256. Integrating our plug-in refinement yields images with richer details and improved semantic coherence compared to the baselines.

Table 8: Checkpoints of models in experiments

| Datasets | URL |
|---|---|
| ImageNet 256×256 guided | `https://openaipublic.blob.core.windows.net/diffusion/jul-2021/256x256_diffusion.pt` |
| LSUN bedroom 256×256 | `https://openaipublic.blob.core.windows.net/diffusion/jul-2021/lsun_bedroom.pt` |
| Stable Diffusion | `https://huggingface.co/CompVis/stable-diffusion-v-1-4-original/resolve/main/sd-v1-4.ckpt` |
| Stable Diffusion v2 | `https://huggingface.co/stabilityai/stable-diffusion-2` |

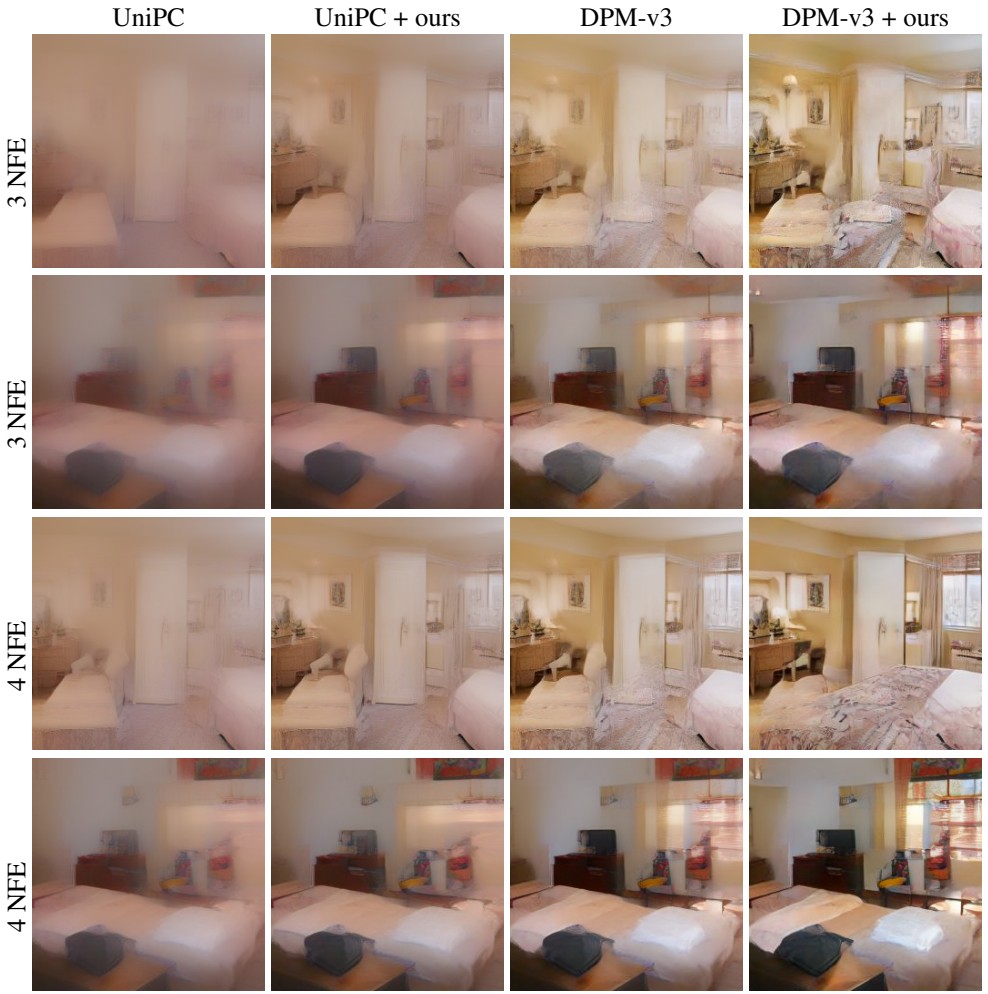

Figure 4: Visualization of generated images using LUSN bedroom 256×256of UniPC, DPM-Solver-v3, and our method integrated with 3 and 4 NFEs. We refer to DPM-Solver-v3 as DPM-v3.

