# OpenReview forum: "Free Lunch at Inference: Test-Time Refinement for Diffusion Models"
_ICLR.cc/2026/Conference — ICLR 2026 Conference Withdrawn Submission_

### Official Review · Reviewer_Rb81 · 2025-10-27

**Soundness:** 2
**Presentation:** 2
**Contribution:** 2
**Rating:** 2
**Confidence:** 3

**Summary:**

This paper proposes a zero-cost test-time refinement method for Diffusion Probabilistic Models (DPMs) during inference. The goal is to enhance the quality of generated images and sampling efficiency without requiring additional training or model evaluations. The authors delve into the geometric characteristics of the latent variable manifold in DPMs, revealing an inherent "isotropic" property derived from their Gaussian process formulation. Building on this insight, they design a refinement step that can be seamlessly integrated into existing samplers. This method is shown to reduce discretization error in sequential sampling methods and accelerate the convergence of parallel sampling strategies, thereby significantly cutting down computational cost and sampling time while maintaining or improving image quality.

**Strengths:**

1. Zero-cost refinement: The most significant advantage is its ability to perform refinement during inference without extra training or model evaluations. This means the method can be directly applied to existing DPMs without retraining, substantially reducing implementation costs and complexity.
2. Broad compatibility: The proposed refinement method can be seamlessly integrated into various existing sequential and parallel sampling frameworks, such as UniPC and DPM-Solver-v3, demonstrating good generality.

**Weaknesses:**

1. The predictor-refiner paradigm doesn't not apply to all the samplers. DDIM and DPM-Solver fall outside this paradigm. This poses challenges to the generality of this mehod.
2. The performance lift is marginal. Table 1.2.3 only achieves limited performance gain compared to the baseline. Adopting one more NFE gets much better gains than this thchnique.
3. The effecitiveness of this method lacks comprehensive evaluation. The experiments on the DiT architecture are missing. The experiments on more other NFEs(besides the 3,4,5 NFEs) are missing. The experiments on more samplers (DDIM, DPM-Solver) are missing.

**Questions:**

Please refer to the weakness part.

---

### Official Review · Reviewer_Q5a1 · 2025-10-30

**Soundness:** 3
**Presentation:** 2
**Contribution:** 2
**Rating:** 4
**Confidence:** 4

**Summary:**

The paper identifies a near-isotropic response of diffusion models’ noise predictor in latent space and exploits it to perform test-time refinement without extra model evaluations.
Specifically, it proposes a plug-in correction that updates the predicted noise using the corrector’s state change within a predictor–corrector step, with a simple scalar gain ($\approx 1/\sigma^2$), yielding a “free lunch” at inference.
The same idea is adapted to parallel/Picard samplers (e.g., ParaDiGMS) via a dual-update that improves convergence per model call.
Analytically, the method is motivated by a local Gaussian/Hessian view of the perturbed marginal $q_t(x)$, which makes the change in predicted noise align with the perturbation direction, justifying the correction.
Empirically, across image generation benchmarks, it improves FID at fixed NFE, with the largest gains at very low step counts.

**Strengths:**

A key strength of the paper is that it articulates a clear geometric insight about diffusion models, showing that the predicted noise responds almost isotropically to small perturbations and supporting this claim with analysis and empirical measurements. Building on this, the authors design a training-free, plug-in refinement that corrects the predicted noise using the corrector’s latent displacement within a predictor–corrector step so no additional model evaluations are required. The idea generalizes to parallel Picard-style samplers via a dual update inside each iteration, which improves convergence per evaluation. Because the technique is sampler-agnostic and orthogonal to solver and schedule design, it composes naturally with modern accelerators rather than competing with them.

Experiments on standard image generation benchmarks show consistent quality gains at fixed step counts, with especially strong benefits when the number of steps is very small. The method integrates with existing pipelines with minimal code changes while preserving compatibility with common guidance strategies. The paper also provides ablations and supplementary analyses that clarify design choices and validate the underlying isotropy assumption.

**Weaknesses:**

### Assumption is a little strong
The method relies on an approximate isotropy assumption and a scalar gain to correct the predicted noise, which means it cannot capture direction-dependent curvature and may over- or under-correct when the corrector’s displacement is misaligned with the true local geometry.

In addition, the method presumes access to a corrector-style update or a Picard window, so integration may be less natural for purely one-step or tightly tuned ODE solvers that do not include a corrector phase. The theoretical motivation leans on local Gaussian or Hessian approximations without quantitative error bounds, so it is unclear how robust the alignment remains in highly non-Gaussian regions or under strong guidance. The empirical scope focuses on standard image generation setups and does not convincingly demonstrate generalization to higher-resolution text-to-image systems, video, audio, or safety-relevant evaluations.

### Marginal performance improvement
The largest benefits appear most plausible in very low step regimes where discretization error dominates, while gains are likely to shrink once strong solvers already achieve low error at moderate step counts.

### Minor comments
- Use parenthetical citation (i.e., \citep) instead of textual citation (i.e., \citet) when you cite papers. \citet should only be used when the authors name(s) are to be read as part of the text.
- log -> \log (lines 174, 192, 193)
- scale score function -> scaled score function (line 224)

**Questions:**

- How robust is the near-isotropy observation across datasets, architectures (pixel vs latent models), and guidance scales, and do you have quantitative thresholds that predict when the assumption begins to break down?
- The refinement uses a scalar gain for the noise update; did you explore simple anisotropic variants (e.g., per-channel or low-rank adaptations) that balance benefit and overhead, and if so what failure modes did you see?
- How does the method interact with different parameterizations (noise, signal, velocity) and with solver families (single-step vs multistep, ODE vs SDE predictor–corrector)?
- What is the computational overhead in wall-clock terms even if there are no extra model evaluations?

---

### Official Review · Reviewer_PqtJ · 2025-10-30

**Soundness:** 4
**Presentation:** 3
**Contribution:** 3
**Rating:** 6
**Confidence:** 4

**Summary:**

The paper used a very elegant approach, leveraging the roughly isotropic property of ideal / learned score field, using this near identity Hessian structure to improve the previous Predictor corrector sampler. Specifically, the near identity Hessian can be used to locally correct the noise prediction via Taylor expansion without neural network evaluation.

Further they showed improved performance on both sequential and parallel solver of diffusion: at few step regime, adding the correction made highly consistent improvement (although sometimes small effect) on UniPC and DPMv3 solver.

It could become a default improvement in the standard PC solver.

**Strengths:**

- The idea and method is very elegant and simple to implement as improvement.
- The algorithmic effect is highly consistent and robust, and tested effective on both sequential and parallel solver case, proving the underlying validity of the idea.

**Weaknesses:**

- Notation clarity about the parameter $\lambda$
    - $\lambda_t$ appears both around Eq.5 defined as half log SNR, and also in Eq. 17, are they the same $\lambda_t$? or the latter is a tunable parameter?
    - Around L385 the authors clarifies they use $1/\sigma_{t-1}^2$ as $\lambda$ in algorithm which makes much more sense. I think the authors could consider redefine the log SNR or $\lambda$ to avoid naming collision.
    - I feel the authors could more explicitly state the expansion of $\epsilon_{t-1}$ with 2nd order info and isotropic approx in the main text, which motivates the algorithm and choice of $\lambda$ better.

**Questions:**

### Questions

- Other line of works which used the intrinsic geometric structure of diffusion trajectory to accelerate sampling may worth mentioning.
For example, similar to authors, [^1] noticed the early phase score field is very much dominated by Gaussian linear score, thus the trajectory can be solved analytically for free, thus could teleport the actual sampling trajectory. I feel their difference from author is that they used a non-isotropic Gaussian (data cov informed) instead of an isotropic one.
[^2] used the low dimensional property of the diffusion models to efficiently correct the sampling trajectory. I feel their difference is that they still requires a bit of learning though here you don’t.

[^1] Wang, B., & Vastola, J. J. (2024). The unreasonable effectiveness of gaussian score approximation for diffusion models and its applications. TMLR

[^2] Wang, G., Peng, W., Li, L., Chen, W., Cai, Y., & Su, S. (2025). Diffusion Sampling Correction via Approximately 10 Parameters. ICML

- One line of work that worth mentioning is [^3], which made the similar observation that the score function / noiser prediction function is “equivariant” — the output move the same way as the input —  in the same sense as the authors say isotropic / identity Hessian. Although they go on different direction and studied adversarial training.

[^3] Rosaria, B. M., Mirza, M. H., Lisanti, G., & Masi, I. (2025). What is Adversarial Training for Diffusion Models?

- The Dual solver just means it update twice, nothing to do with dual space?
- It’s a little bit out of scope, but if the authors want to add a bit learning to the sampler to squeeze more juice out of the idea, what would they learn / customize?

---

### Official Review · Reviewer_Jk3h · 2025-10-31

**Soundness:** 2
**Presentation:** 2
**Contribution:** 2
**Rating:** 2
**Confidence:** 4

**Summary:**

Diffusion models work well, but can be expensive to sample from, since sampling requires running a multi-step reverse process. Usually, research on speeding up sampling concentrates on developing efficient numerical methods for integrating ODEs and SDEs (think, e.g., RK4 versus Euler steps); this type of work exploits the fact that the reverse process can be cast as a system of ODEs/SDEs. In this work, the authors discuss an alternative "geometric" approach that exploits Gaussian structure in the reverse process.

Their insight is that, for 'most' of sampling, the sampling distribution remains fairly isotropic. Given this fact, the authors propose to improve predictor-corrector sampling schemes by approximately correcting the update to use a potentially better estimate for $x_t$ (that is, to use the corrected $x^c_t$ instead of $x_t$). They show empirically that this change seems to improve the quality of generated samples.

**Strengths:**

The idea of refining predictor-corrector sampling schemes by using a better choice of $x^c_t$ is generally interesting, and it is nice that the authors' specific choice seems to improve performance.

The idea of exploiting the Gaussian structure of the sampling distribution for $t$ close to $T$ is good, and has been used in a lot of prior work.

**Weaknesses:**

Overall, I found this paper kind of difficult to understand. It is kind of confusingly written, and more importantly the proposed method seems only dubiously supported.

**Theoretical issues.** If I understand the key insight correctly, the authors claim that, early in sampling (i.e., early in the reverse process), the sampling distribution is approximately isotropic. This is absolutely true; the initial condition is a Gaussian, and it remains Gaussian for a large range of $t$. As the authors state, only for relative low $t$ / noise scales, anisotropy comes in.

I think my two fundamental theoretical issues are as follows. First, it is kind of relative what one means by "most" of sampling. In terms of $t$, it is indeed true that the sampling distribution is isotropic for "most" of sampling. But many noise schedules used in practice spend many more steps near very small values of $t$, since this can improve performance. This is true, for example, of the schedules recommended by the Karras et al 2022 EDM paper (https://arxiv.org/abs/2206.00364). For such schedules, because more steps are spent near $t = 0$, it is not true that the sampling distribution is isotropic for 'most' steps.

Second, I'm having a hard time connecting this to other similar work. One work this paper makes me think of is the Wang and Vastola 2024 TMLR paper (https://arxiv.org/abs/2412.09726). In it, the authors propose that, since empirically the sampling distribution is Gaussian at large times $t$, one can use the analytic solution to the reverse process for Gaussian scores to immediately 'teleport' from time $t = T$ to some $t = t' << T$, without doing any numerical steps at all.

The theoretical insight of this paper strikes me as the same thing: if the method works, it's because for large times the sampling distribution is approximately Gaussian. But in that case, why not just use the Gaussian solution and do the teleportation thing? Why do this predictor-corrector scheme? It would be interesting if the authors could compare a teleportation-based approach to their proposed method.

A final note on a theoretical justification for the authors' idea: I guess I'm just not understanding why this method should work. Is it mainly that, for large $t$, the sampling distribution is approximately Gaussian? Or is something more subtle going on, such that even for smaller noise scales (where most noise schedules concentrate steps) this still kind of works? It would be very helpful if the authors could do some simple theory to clarify why their method works.

**Empirical issues.** I am also not totally convinced that the method empirically works well. The largest differences happen for very small NFEs (e.g., Table 1, UniPC, NFE = 3), where quality is bad regardless of the use of the authors' method. This is not even universally true; for ImageNet, the improvement seems to be very small. But for larger NFEs, the improvement seems modest.

Can the authors show more impressive speedups/improvements? Maybe I am not looking at the tables carefully enough, but the results in the paper don't seem like they provide a strong motivation to use the authors' method. If more impressive improvements are not possible, the authors should explain why their method is nonetheless useful (e.g., maybe it mostly increases speed while keeping quality similar, or maybe the point is that it's a small improvement, but it's cheap, so why not do it?).

**Other issues.**
- I found the structure of the paper kind of confusing. A lot of time is spent on intro things (Sec. 2 repeats a lot of what was already in the intro; Sec. 3 spends too much space reviewing things people in the field know well), and much of this could probably be trimmed or moved to SI. The authors' core insight, and what they mean by a "geometric" approach to speeding up sampling, isn't really explained until page 5. I think the authors could at least provide some intuition much earlier in the paper. There should be more discussion of the strengths/weaknesses of the method after/during the experiments section; there is currently just a short conclusion.
- Line 50, "As a result, the deployment of DPMs is limited on computationally constrained platforms, restricting their broader adoption." What platforms do the authors have in mind? I can kind of imagine, but this part wasn't totally clear to me.
- In the intro, the authors claim their approach is very different from a 'numerical methods for ODEs/SDEs' approach. But it's not clear to me that this is true. Maybe there is a numerical methods way to view their method.
- The authors should look carefully for typos. See, e.g.,  line 691 "bayes" -> Bayes' ;
line 915 "265of" -> 265 of
- There should be parentheses around most citations, e.g., "Bla bla (Author et al 20XX)" instead of "Bla bla Author et al 20XX". This can be fixed using the \citep command instead of \cite.

**Questions:**

My most important questions are these:

1. What is the core insight of the method? Is it that the sampling distribution is approximately Gaussian early in sampling? If yes, how does their idea compare with the Wang and Vastola 2024 TMLR 'teleportation' idea? Is this approach to speeding up sampling more performant/robust than that one? If so, can the authors show this?

2. What are the typical performance improvements one can expect by using this method? (e.g., for 10 NFEs, on these data sets, one gets an accuracy improvement of X and a speedup of Y) Does the method still work if the noise schedule spends most steps on small values of $t$?

---

### Note · Authors · 2025-11-14

**Comment:**

We sincerely thank the reviewers for their time and feedback.

**Withdrawal Confirmation:**

I have read and agree with the venue's withdrawal policy on behalf of myself and my co-authors.